# Marine Copepods as a Microbiome Hotspot: Revealing Their Interactions and Biotechnological Applications

**Jiantong Feng** [1,2,3,4], **Maurizio Mazzei** [2], **Simona Di Gregorio** [5], **Luca Niccolini** [1,5], **Valentina Vitiello** [1,4], **Yingying Ye** [3,4], **Baoying Guo** [3,4], **Xiaojun Yan** [3,4] **and Isabella Buttino** [1,4,*]

1   Italian Institute for Environmental Protection and Research, ISPRA, Via del Cedro 38, 57123 Livorno, Italy; jiantong.feng@phd.unipi.it (J.F.); luca.niccolini95@gmail.com (L.N.); valentina.vitiello@isprambiente.it (V.V.)
2   Department of Veterinary Sciences, University of Pisa, Viale delle Piagge 2, 56124 Pisa, Italy; maurizio.mazzei@unipi.it
3   National Engineering Research Center for Marine Aquaculture, Zhejiang Ocean University, Haida South Road 1, Zhoushan 316022, China; yeyy@zjou.edu.cn (Y.Y.); yanxj@zjou.edu.cn (X.Y.)
4   Functional Biology of Marine Biota, Sino-Italian Joint Laboratory ZJOU-PRC and ISPRA-Italy, Via del Cedro 38, 57123 Livorno, Italy
5   Department of Biology, University of Pisa, Porta Buozzi 1, 56126 Pisa, Italy; simona.digregorio@unipi.it
*   Correspondence: isabella.buttino@isprambiente.it

**Abstract:** Copepods are the most abundant organisms in marine zooplankton and the primary components of the food chain. They are hotspots for highly adaptable microorganisms, which are pivotal in biogeochemical cycles. The microbiome, encompassing microorganisms within and surrounding marine planktonic organisms, holds considerable potential for biotechnological advancements. Despite marine microbiome research interests expanding, our understanding of the ecological interactions between microbiome and copepods remains limited. This review intends to give an overview of the recent studies regarding the microbiome associated with marine copepods, with particular focus on the diversity of bacteria and fungi. The significance of copepod-associated microbiomes in different contexts, such as aquaculture and biodegradation processes, was evaluated. The ability of the microbiome to mitigate harmful bacterial growth in cultured organisms was also explored. The microbiome associated with copepods has demonstrated efficacy in reducing the proliferation of detrimental bacteria in aquaculture, paving the way for the commercial utilization of natural zooplankton in fish rearing. Additionally, copepod-associated microbiomes may play a role in addressing marine environmental challenges, such as the bioremediation of polluted marine matrices. Overall, this review represents a basis for investigating intricate copepod-associated microbiomes and their diverse applications, enhancing our comprehension of the ecological and evolutionary significance of marine microbiomes.

**Keywords:** zooplankton; microorganisms; bacteria; fungi; aquaculture; biodegradation

## 1. Introduction

Plankton, including both phyto- and zooplanktonic organisms, are at the base of trophic webs in all aquatic ecosystems and contribute significantly to the biodiversity of marine ecosystems [1,2]. These organisms confer ecological benefits to marine environments; other than functioning as a crucial food source for many organisms, they participate in the removal of carbon dioxide from the atmosphere, generating oxygen [3]. Additionally, they contribute to the decomposition of deceased plants and animals, leading to the sequestration of both temporary and permanent carbon in the deep ocean. They participate in this carbon sequestration process alongside the microbial carbon pump, which involves mechanisms like the conversion of organic matter into dissolved inorganic carbon (DIC) and the subsequent temporary storage of this carbon in deeper waters until it is resurfaced through thermohaline circulation (Figure 1).

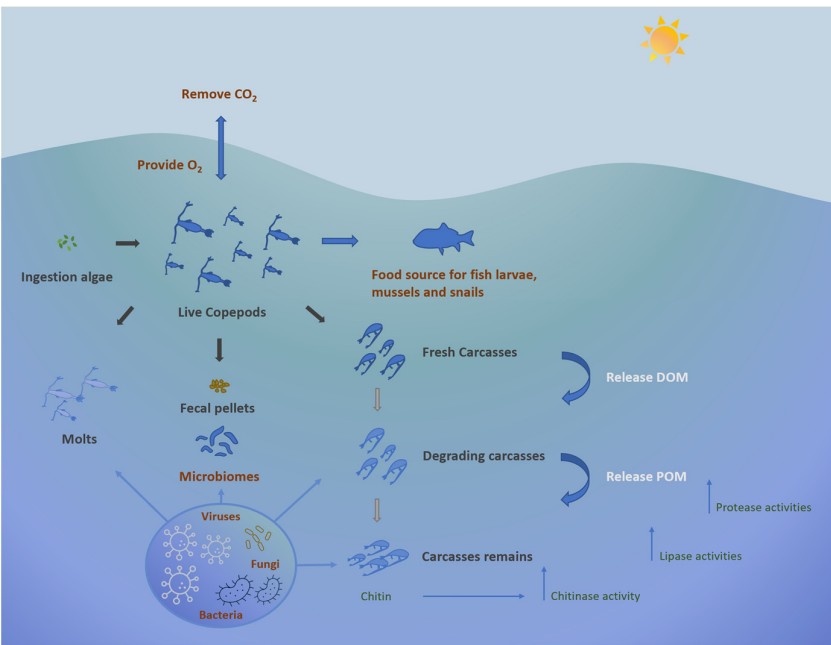

**Figure 1.** Ecological role of copepods in marine environment.

Among zooplankton, copepods, a class within the Crustacean subphylum, predominate as the most abundant multicellular organisms on Earth, potentially comprising 80% of the biomass of medium-sized zooplankton [4–6]. Current knowledge identifies more than 11,300 copepod species, showcasing their remarkable diversity and wide distribution across aquatic environments while also occurring in moist terrestrial habitats [7–9]. Copepods can be benthic-pelagic organisms, inhabiting the water column and sediment, and even include parasites [10,11]. Moreover, marine copepods play a critical role in the microbial loop, significantly contributing to the cycling of dissolved organic material (DOM) and microelements through their feeding activities [6,12,13]. Copepods also contribute to the release of dissolved organic carbon (DOC) and particulate organic carbon (POC) during feeding and excretion, thus providing nutrients for associated microorganisms [14,15]. Collectively, copepods regulate microbial populations, recycle nutrients, and facilitate the transfer of energy within the food web. Due to their feeding habits, nutrient excretion, and vertical migration, copepods contribute to the overall productivity and function of aquatic ecosystems and are keystone organisms in marine food webs.

Due to the position of copepods in marine food chains, they are an important alternative to traditional live feed in aquaculture. In fact, in comparison to more conventional feeds such as *Artemia* and Rotifers, copepods demonstrate superior capacity in ensuring nutritional quality and enhancing the digestibility of food for animals [16]. Utilizing copepods in aquaculture offers several advantages, including improved growth, survival, tolerance, and a reduced incidence of deformities in fish larvae [17,18]. Extensive research has been dedicated to exploring the benefits of copepods in aquaculture [19–22], and the observed enhancements in fish larvae performance are also due to the microbiome associated with copepod cultures [23].

In recent years, symbiotic interactions in crustacean arthropods, such as copepods, have gained significant attention [24]. The microbiome associated with copepods is a key factor in regulating the overall energy balance of zooplankton, ensuring homeostasis, and influencing the cycling of organic matter in aquatic ecosystems [25,26]. Moreover, the gut microbiota of copepods acts as a driving force for their adaptation and acclimation to harmful cyanobacterial blooms by facilitating the degradation of toxic substances released by cyanobacteria [27]. Furthermore, microbial communities associated with copepod carcasses participate in the metabolic processing of nutrient uptake, including denitrification, particularly in anoxic environments [28,29]. In this review, we provide an overview of the most

recent scientific advances related to the study of the copepod-associated microbiome and discuss future biotechnological applications in bioremediation of environmental matrices and to improve aquaculture production.

## 2. Characterization of the Copepod-Associated Microbiome

Identifying microbiomes associated with aquatic organisms involves a combination of techniques to characterize the microbial communities living in and on these organisms. These techniques help to understand the diversity, composition, and functionality of the microbiota. Prior to the advent of molecular techniques, the identification and characterization of microbiota in environmental samples involved a culture-dependent approach. The method relies on growing microorganisms in a laboratory setting, usually on culture media, to isolate and identify them. Microorganisms were grown in isolated conditions under controlled conditions of temperature, pH, and oxygen [30]. This method allows the isolation and pure culture of specific microorganisms, providing the opportunity to study the physiology, metabolism, and characteristics of individual microorganisms. Microorganisms were identified through techniques such as microscopy, biochemical testing, and genetic sequencing. However, the culturomics approach has limitations due to the specific growth requirements of different organisms, and conventional microbiological techniques only capture a small fraction of the microbiota present in environmental samples. The process is time-consuming, as some microorganisms can take days or even weeks to grow. Additionally, laboratory culture conditions may not accurately represent the natural environment; they may overlook the most abundant microorganisms in the environment, including those that are not easily cultivable, leading to bias in the types of microorganisms that can be isolated [31].

Culture-independent molecular techniques, such as next-generation sequencing (NGS) and meta-barcoding, have advanced our understanding of microbial diversity in marine and freshwater environments [32]. It is currently the most used method to analyze the microbiome, designed to study microorganisms without requiring their culture.

Analysis of genetic material directly from environmental samples can allow the identification and characterization of microorganisms [33]. It provides a more comprehensive view of the microbial community, as it can detect unculturable and rare microorganisms, allowing the study of the genetic content and biodiversity of entire microbiomes [34].

The advancement of genome sequencing technologies has revolutionized the field of microbiome research, with many studies employing Illumina short-read and nanopore long-read technologies for sequencing [35,36]. Illumina next-generation sequencing is a DNA sequencing technology used to determine the order of base pairs in DNA [37]. Currently, the most commonly used method for analyzing microbial communities associated with aquatic organisms is high-throughput sequencing of the 16S rRNA gene, predominantly performed on the Illumina platform [38–40]. The 16S rRNA gene contains nine hypervariable regions with distinct characteristics, with the V3 and V4 regions being the most frequently sequenced [41]. This technology can be utilized for various applications, such as whole genome and fragment sequencing, metagenomics, transcriptome analysis, and methylation analysis [42]. The process involves three fundamental steps: amplification, sequencing, and analysis [43].

This technique is faster and more efficient than a culture-dependent method. Its disadvantage is that it does not provide information on the physiological and metabolic characteristics of individual microorganisms and requires specialized equipment and expertise for genetic analysis. The functional roles of the copepod-associated microbiome are also being revealed [9]. For instance, Moisander et al. [39] employed 16S rRNA amplicon sequencing to identify a high abundance of the Proteobacteria phylum in copepods, along with Planctomycetes. Shoemaker and Moisander [44] highlighted the importance of the copepod gut microbiome on the function of bacterioplankton in seawater, suggesting that despite low microbial abundance, copepods can contribute to bacterioplankton function across diverse oceanic regions. Yeh et al. [45] utilized 16S rDNA metabarcoding to ana-

lyze the eukaryotic and prokaryotic diversity in the gut contents of the copepod *Calanus finmarchicus* in the North Atlantic Ocean. This approach provided insights into the diet, microbiome, parasites, and pathogens of copepods. Notably, while *Vibrio* spp. Was commonly observed in culture-dependent studies, it was relatively rare in high-throughput sequencing analyses [9]. Overall, the combination of these techniques provides a more comprehensive understanding of the microbiome and its ecological role associated with aquatic life and is tailored to the specific research goals and the nature of the microbial community being studied.

## 3. Copepod-Microbiome Association

Over the past decade, with the proliferation of various new molecular techniques such as DNA sequencing and metagenomics, researchers have increasingly explored the marine microbiome associated with copepods, enabling more detailed characterization of taxonomic composition and diversity (Table 1). Gerdts et al. [46] selected the bacterial communities associated with four prevalent copepod species, namely *Acartia* sp., *Temora longicornis*, *Centropages* sp., and *Calanus helgolandicus*, obtained from the North Sea. Denaturing gradient gel electrophoresis (DGGE) and 16S rDNA fragment sequencing methods were employed to examine the overall bacterial community composition and its seasonal dynamics. As said above, this molecular approach facilitated the determination of bacterial phylogenetic positioning, independent of ecosystem complexity and culturability, thereby revealing a relatively low bacterial diversity. Analyzing the DGGE banding pattern of the clone library over a two-year period, no significant differences were observed between copepod species communities or across seasons. Within the bacterial phyla Actinobacteria, Firmicutes, Proteobacteria, and Bacteroidetes, Alphaproteobacteria exhibited the highest abundance, with Gammaproteobacteria dominating in PCR-DGGE and clone libraries, while Alphaproteobacteria predominated in PCR-DGGE analysis.

Bickel et al. [47] employed a similar molecular approach to study the bacterial communities associated with copepods in the York River, along with the free-living communities in the river, with the aim of assessing differences in composition and function over time. Analysis of the genetic fingerprints of bacterial communities and their utilization of carbon substrates revealed that copepod-associated bacterial communities exhibited distinct genetic profiles compared to free-living communities, although they shared similar arrays of carbon substrates. Notably, bacteria associated with different zooplankton groups exhibited greater genetic similarity during the same period. Furthermore, the microbial environment associated with plankton demonstrated greater stability compared to the free water. Seasonal variations in environmental conditions were also identified as important factors influencing the composition and function of bacterial communities. Exploring the microenvironment of copepods in the water column could provide a more comprehensive assessment of bacterial abundance, overall function, and biodiversity.

De Corte et al. [38] conducted a comparative analysis of the in vitro and in vivo habitats of bacteria associated with copepods in the North Atlantic Ocean. They investigated the bacterial composition using 454 high-throughput 16S rRNA gene sequencing. Significant differences were observed between bacterial communities associated with ambient water and copepod families, specifically Centropagidae and Clausocalanidae among Calanoida and Corycaeidae, Oncaeidae, and Lubbockiidae among Cyclopoida. The copepod-associated communities were predominantly dominated by Bacilli and Actinobacteria, while the free-living communities were dominated by Synechococcus, Alphaproteobacteria, and Deltaproteobacteria. This indicates a dynamic connection between bacteria in the water environment and copepods, which influences the activity and diversity of copepod-associated bacteria. The authors suggested that the diet of copepods may also impact the composition of their gut-associated bacterial populations.

Moisander et al. [39] used high-throughput sequencing research to characterize the microbiomes associated with the copepods *Acartia longiremis*, *Centropages hamatus*, and *C. finmarchicus* during early summer in temperate regions. Some bacterial taxa belonging to

Gammaproteobacteria showed a stable association with copepods, independent of their feeding habits. On the other hand, copepods can actively adjust feeding rates and prey selection based on the availability and quality of food resources in their environment. This ability to feed helps copepods optimize nutritional intake and maximize energy acquisition. The results indicated that bacteria, such as *Pseudoalteromonas* and *Vibrio* spp., showed stable association with copepods in the Gulf of Maine, becoming more abundant later in the summer as the water temperature increased.

Dorosz et al. [40] utilized high-throughput methods to investigate the microbiomes of the Danish neritic copepod species *Acartia tonsa* and *Temora longicornis*. They found significant differences between the two copepod species in the same environment in terms of bacteria composition and relative abundance; Alphaproteobacteria were predominantly found in the *A. tonsa* microbiome, while Gammaproteobacteria were the most abundant bacteria associated with *T. longicornis*. These findings supplemented our understanding of specific copepod-associated bacteria and indicated potential differences in microbiomes among different copepod species in the same environment.

Copepods serve as a natural food source for fish larvae in larviculture and can also act as a vehicle for probiotics. Zidour et al. [48] isolated bacteria from the copepod *A. tonsa* eggs and used high-throughput identification via MALDI-TOF analysis. The identified bacteria included *Vibrio*, which may be potentially pathogenic to both fish and humans, as well as *Staphylococcus*, *Pseudomonas*, and the beneficial genus *Bacillus* for both fish and humans. Meanwhile, two antagonistic experiments were carried out against the *Bacillus pumilus* and *Bacillus subtilis* strains. Further analysis revealed that the antagonistic activity of *B. pumilus* may be caused by compounds from the amicoumacin family, which is known for its potential to inhibit bacterial growth.

Shoemaker and Moisander [44] conducted a survey of copepod gut microbiomes at the Bermuda Atlantic Subtropical Gyre. They obtained copepod guts using sterile needles under a dissecting microscope and analyzed the gut bacterial microbiome of copepods through bacterial amplification sequencing. Persistent bacterial groups such as Actinobacteria, Bacteroidetes, and Firmicutes were consistently present in the copepod gut throughout the year, exhibiting synchronous changes over time. The gut communities exhibited clear differences from those present in the surrounding seawater, highlighting that copepod guts host unique and specialized microbial communities over extended periods. Furthermore, copepod intestinal bacteria showed significant changes during spring and early winter, with the relative abundance of *Synechococcus* being highest during these seasons. Anaerobic bacteria, such as Clostridiales, were found to inhabit dynamic microhabitats in low-nutrient open seas. In a study by Shoemaker et al. [49], culture experiments on copepod bacterial communities in the North Atlantic Subtropical Gyre demonstrated that copepods promote the growth of Rhodobacteraceae, Vibrionaceae, and Oceanospirillales in the surrounding waters. Additionally, copepods harbor and release their own specific bacterial groups, Flavobacteriaceae and Pseudoalteromonadaceae, either in or on their bodies. Therefore, the copepod microbiome has the potential to influence marine bacterial biomass and exert control over the distribution and composition of bacterial communities in seawater.

Chae et al. [50] examined the gut bacterial communities of three dominant copepod species, *Acartia hudsonica*, *Sinocalanus tenellus*, and *Pseudodiaptomus inopinus*, in a brackish reservoir and investigated their variation based on copepod species and environmental conditions to elucidate the mechanism of their interactions with phytoplankton and bacteria. The study revealed differences in the gut bacterial communities among these copepod species, with the core bacteria belonging to the Rhodobacteraceae family, which is known for its abundance in both marine and freshwater ecosystems. The study found that certain core bacteria were consistently present across copepod species and locations, including *Novosphingobium capsulatum* and the family Rhodobacteraceae. These core species are known for their ability to decompose organic substrates, demonstrating their importance in the digestive processes of copepods. Moreover, the bacterial community of copepods

exhibits high variability and is greatly affected by environmental factors, particularly salinity; lower salinity levels were associated with greater changes in the composition of the bacterial community in *P. inopinus*. These results indicated that the gut microbiota of copepods was influenced by both the environment and their feeding behavior. Overall, the study highlighted that the gut bacterial communities of different copepod species responded differently to environmental variables, such as seasonality, growth location, and salinity, as well as copepod feeding behavior.

### 3.1. Copepod-Bacteria Interactions

Plankton are peculiar microhabitats for bacterial communities in aquatic ecosystems [51]. On the other hand, bacterial communities play a pivotal role in copepod physiology, contributing to the dynamics of the food web and biogeochemical cycling [44]. The relationship between copepods and bacteria is basically distinguished into two main types: endobiotic and exobiotic associations.

### 3.1.1. Endobiotic and Exobiotic

Endobiotic associations involve bacteria that inhabit the internal organs, tissues, or digestive tracts of copepods, represented by copepod gut bacteria. These bacteria can establish symbiotic relationships with copepods. Some endobiotic bacteria can also assist copepods in digesting complex organic matter, such as phytoplankton or detritus, by breaking down these substances into more easily digestible forms. In return, copepods provide a protected and nutrient-rich environment for these bacteria. These bacteria help digest the complex carbohydrates and cellulose present in phytoplankton, making them an easier food source for copepods [52]. These associations may differ between copepod species, environmental conditions, and life stages. Furthermore, the distinction between two associations is not always strict, as some bacteria may switch between internal and external habitats at different stages of their life cycle. It is critical for studying copepod ecology, their role in marine food webs, and their interactions with microbial communities in aquatic ecosystems.

Studies on copepod-bacteria associations have revealed that copepods are primarily linked with taxa such as Bacteroidetes, Alphaproteobacteria, and Pseudoalteromonadaceae [40,53]. The composition of the microbial ecology of the copepod guts varied significantly across different regions in the North Atlantic Ocean, shedding light on the differences in food webs within pelagic ecosystems.

Bacterial communities inhabiting nutrient-rich environments are likely to contribute to various biogeochemical cycles in the ocean, including methanogenesis, denitrification, mercury methylation formation, iron remineralization, and organic compound degradation [54]. For example, certain bacteria associated with copepods can facilitate the conversion of organic carbon to carbon dioxide through respiration, influencing carbon cycling in the ecosystem [55]. Additionally, some microorganisms can participate in the cycling of other elements like sulfur or iron, impacting the availability and transformation of these elements in the environment [56,57]. Moreover, they may interact with contaminants present in the water, influencing their fate and effects. Some microorganisms can degrade or transform pollutants, contributing to the bioremediation of contaminated environments [58]. Conversely, certain contaminants, such as heavy metals or pesticides, may affect the composition and activity of copepod-associated microbial communities, potentially altering their ecological functions [59].

The bacterium *Wolbachia* has been observed to induce feminization in host copepods. *Mesocyclops aspericornis* and *M. thermocyclopoides* transmitted it to their offspring. Wiwatanaratanabutr [60] conducted the first comprehensive surveys of *Wolbachia* infections in cladocerans and copepods from various regions in Thailand. The maternally inherited *Wolbachia* bacteria were found to induce reproductive alterations in copepods. The density of bacterial infection varied among copepod species, with higher levels observed in *M. thermocyclopoides*.

Endobionts living within copepods may compete with their hosts for essential nutrients [61]. This competition can lead to reduced nutrient absorption by the copepod, potentially affecting its growth, reproduction, and overall fitness. Some endobionts can interfere with the reproductive capabilities of copepods. For example, parasitic endobionts may divert energy and resources away from copepod reproduction, leading to reduced fecundity and population growth [62]. They can suppress the immune system of the host or induce chronic stress responses, making copepods more susceptible to infections and diseases, both from the endobionts themselves and from external pathogens. Furthermore, certain endobionts produce toxins or metabolic byproducts that can harm copepod tissues and organs, potentially causing physical damage or impairing essential physiological processes. Some endobionts can interfere with copepod behavior, potentially making them more vulnerable to predation by disrupting their ability to escape from predators or find suitable food sources. Overall, the negative effects of endobiotics can lead to reduced fitness and survival of individual copepods, which can, in turn, impact copepod populations and their role in aquatic food webs and nutrient cycling.

In addition to parasitic interactions between copepods and bacteria, commensalism was also described [63]. The prokaryote communities associated with copepod guts may provide metabolic benefits to copepods. Tang [64] investigated the role of copepods as microbial hotspots in the ocean by studying the effects of copepod feeding activities on associated bacteria. The study found a balance between bacteria growth stimulated by copepod feeding and bacteria loss through copepod defecation. The bacterial population associated with copepods was significantly higher than that of marine free-living bacteria. Time-series experiments showed an increase in bacterial abundance inside the bodies of copepods after feeding, with a higher growth rate than free-living bacteria. Copepods released a significant number of bacteria through defecation when food was present, leading to high bacterial abundance in fecal pellets. Different diets influenced the growth kinetics of bacteria associated with copepods, suggesting that diet types may shape bacterial communities within copepod hosts.

The gut of copepods plays a crucial role in providing a microaerophilic or hypoxic microenvironment, even in small copepods. This unique physiological characteristic of the copepod gut creates an environment with low oxygen levels, which is conducive to the survival and growth of specific microorganisms, including bacteria [65,66]. These bacteria, residing within the copepod gut, make significant nutritional contributions to copepods. Within the copepod gut, bacteria can perform various functions that benefit the host. They can assist in the breakdown and digestion of complex organic matter, such as detritus and phytoplankton, into simpler and more easily assimilable forms [67]. Through processes such as fermentation and enzymatic degradation, these bacteria can release essential nutrients and energy sources that copepods can readily utilize for their growth, reproduction, and survival. Moreover, the presence of bacteria in the copepod gut can enhance the ability of copepods to extract nutrients from their food.

Bacteria can produce enzymes that copepods themselves may lack, allowing them to access and utilize specific nutrients that would otherwise be inaccessible or difficult to obtain [64]. This symbiotic relationship between copepods and bacteria ensures that copepods can maximize their nutrient uptake efficiency and optimize their nutritional status. Additionally, bacteria in the copepod gut can provide other benefits to copepods, such as the synthesis of essential vitamins and cofactors that may be lacking in their diet [27]. These micronutrients play vital roles in various physiological processes, including metabolism, reproduction, and immune function. By supplying copepods with these essential micronutrients, the bacteria contribute to the overall health and fitness of the copepod population.

In summary, the copepod gut serves as a specialized microenvironment with reduced oxygen levels, creating favorable conditions for the presence and activity of bacteria. This intricate relationship between copepods and bacteria highlights the significance of microbial

interactions within the copepod gut and their impact on copepod ecology and ecosystem functioning [68,69].

In exobiotic association, bacteria live on the outer surface of the copepod, including their chitinous exoskeletons and appendages, with beneficial or commensal relationships [70]. Beneficial bacteria can act by deterring harmful pathogens or assisting in the own defenses of copepods. Moreover, some exobiotic bacteria can produce antimicrobial compounds that help protect copepods from harmful pathogens, parasites, or fouling organisms. For example, studies have shown that copepods (*Calanus* spp.) harbor specific bacteria on their exoskeletons. These bacteria produce compounds that inhibit the attachment and growth of fouling organisms, helping to keep copepod surfaces clean [71].

In terms of copepod interactions with exobiotic associations, certain bacteria exhibit cleansing properties that impede fouling by minimizing the accumulation of debris in the surface recesses of copepods, consequently enhancing their swimming capabilities [71]. Bacterial colonization of copepod exoskeletons, specifically *Acinetobacter*, *Bacillus*, and *Flavobacterium*, correlates with the substantial accumulation of phytoplankton food debris between appendages and in abdominal crevices [72].

Copepod-associated microbiomes also have various environmental benefits. Microbiomes play a role in nutrient cycling within aquatic environments, contributing to the decomposition of organic matter and releasing essential nutrients such as nitrogen and phosphorus into water. This recycling of nutrients is crucial for the growth and productivity of primary producers like phytoplankton, which form the base of the aquatic food chain. They can impact energy transfer within the ecosystem. As copepods feed on phytoplankton or other organic matter, they transfer energy from primary producers to higher trophic levels, such as fish larvae or other predators.

Understanding the environmental impact of copepod-associated microorganisms is essential for comprehending the ecological processes and functions of aquatic ecosystems. Further research is needed to elucidate the specific roles and contributions of different microbial taxa associated with copepods and their implications for ecosystem dynamics, resilience, and response to environmental changes [65,67,73,74]. Additionally, the organic matter derived from decomposed fecal pellets satisfies the carbon requirements of thriving planktonic bacteria. Although most ingested bacteria are digested and assimilated by the host, a portion of bacteria remains viable within the intestine and proliferate within the feces. This leads to heightened aminopeptidase activity in the fecal pellet, which plays a role in copepod nutrition and the degradation of organic matter [75].

### 3.1.2. Pathogen Bacteria Associated with Copepods

Some copepod-associated microorganisms can act as pathogens, causing diseases in copepods or other organisms in the ecosystem. These pathogens may disrupt the population dynamics of copepods and other species, leading to changes in community structure and ecosystem functioning. Disease outbreaks among copepods can have cascading effects on the entire food web, affecting the abundance and distribution of other organisms in the ecosystem. In the past, studies have focused on the identification of potential human pathogenic bacteria of the genus *Vibrio* associated with copepods. Kaneko and Colwell [76] found that *Vibrio parahaemolyticus* adheres to chitin particles and copepods, with adsorption efficiency influenced by pH and the concentration of NaCl and other ions in seawater. The highest adsorption efficiency was observed in Chesapeake Bay water samples, while the lowest was in open sea samples. Had higher adsorption efficiency onto chitin compared to other bacterial strains tested. This adsorption effect plays a significant role in the distribution and annual cycle of *V. parahaemolyticus* in estuarine systems.

Huq et al. [77] investigated the ecological relationship between *V. cholerae* and copepods. They discovered that both serogroup O1 and non-O1 strains attach to the surfaces of live copepods collected from Chesapeake Bay and Bangladesh. Scanning electron microscopy revealed the specificity of *V. cholerae* attachment, primarily in the oral cavity and egg sacs of copepods. The presence of live copepods extended *Vibrio* survival time in

the water. Other bacterial strains, such as *Pseudomonas fluorescens* and *Escherichia coli*, did not adhere to live or dead copepods. The attachment of *V. cholerae* to live copepods is an important ecological factor and a key aspect of cholera epidemiology, considering that *V. cholerae* serogroup O1 is the causative agent of cholera. The persistence of the human pathogen *V. cholerae* in the water is prolonged when live copepods are present, whereas this phenomenon is not observed when the bacteria are attached to deceased copepods [71].

Huq et al. [78] examined the impact of water temperature, salinity, and pH on the survival and growth of *V. cholerae* serovar O1 associated with live copepods. They found that higher water temperatures and alkaline pH favored the growth and attachment of *V. cholerae*, while the maximum growth and attachment occurred at a salinity of 15%. These laboratory findings suggest that the attachment of *V. cholerae* to live copepods may also occur in natural estuarine environments, emphasizing its importance in cholera epidemiology.

Araújo et al. [79] examined the influence of the copepod *Mesocyclops longisetus* on the survival of *V. cholerae* O1 serovar Inaba in freshwater. The study confirmed a clear association between *V. cholerae* O1 and live copepods, as the bacteria survived at compatible levels with the initial inoculation for six days in the presence of copepods. More recently, Thomas et al. [80] investigated the salinity-induced survival strategy of *V. cholerae* associated with copepods in the Cochin backwaters. They found that *V. cholerae* occurred in both culturable and non-culturable forms in the tropical estuary. During high salinity periods, *V. cholerae* was associated with copepods in a non-culturable form, but with lower salinity during the monsoon season, the copepod-associated *V. cholerae* reverted to a culturable form. Rawlings et al. [81] explored the association of *V. cholerae* O1 El Tor and O139 Bengal with the copepods *A. tonsa* and *Eurytemora affinis*. The study aimed to understand the colonization patterns of the two serogroups in different copepod species. *Vibrio cholerae* O1 showed higher colonization rates than *V. cholerae* O139 in both adult copepods and multiple life stages of *E. affinis*. These findings suggest that the preferential colonization of copepods by *V. cholerae* O1 may contribute to its predominance in cholera epidemics in rural Bangladesh. De Magny et al. [82] investigated the role of zooplankton diversity in *V. cholerae* population dynamics and cholera incidence in the Bangladesh Sundarbans. They found that *V. cholerae*, including serogroups O1 and O139, were associated with crustacean zooplankton, particularly copepods, in ponds, rivers, and estuarine systems. The study analyzed chitinous zooplankton communities and identified rotifers, cladocerans, and copepods as the dominant groups associated with *V. cholerae* detection and cholera human infection. Local ecological factors were found to influence the interaction between *V. cholerae*, zooplankton hosts, and cholera incidence. These studies highlight the important role of copepods and other zooplankton in the ecology, population dynamics, and survival strategies of *V. cholerae* in different aquatic environments.

Almada and Tarrant [83] conducted transcriptional analysis, which revealed that the attachment of *Vibrio* sp. F10 led to alterations in the expression of copepod host genes associated with immune responses. Furthermore, *Vibrio* spp. Exhibited a greater tendency to attach to the copepod exoskeleton compared to *Escherichia* and *Pseudomonas*. The attachment of copepods induced changes in *Vibrio* cultivability, indicating that copepods are not passive environmental vectors but selectively interact with *Vibrio*. This selective interaction may regulate the abundance and activity of bacteria attached to copepods and their transmission to humans.

### 3.2. Copepod-Fungi Interactions

Copepod-fungal interactions can have positive and negative effects on both copepods and fungi, affecting ecological dynamics and ecosystem processes. Positive interactions such as nutrient cycling, where the fungi can break down complex organic matter, including dead copepods, contribute to nutrient cycling in aquatic ecosystems [84,85]. This decomposition process can release essential nutrients back into the environment, benefiting other organisms.

Some copepods establish reciprocal relationships with fungi [86]. For example, copepods can act as dispersers of fungal spores, helping the fungus to reproduce and spread. On the other hand, fungi can provide nutritional benefits or protection to copepods. Furthermore, mutualistic relationships between copepods and certain fungi can promote the detoxification of harmful substances present in the environment. This can help copepods tolerate other toxic conditions. Some copepods have symbiotic fungi that help digest complex organic matter which helps them obtain nutrients from food sources more efficiently.

The mechanisms of host-parasite interactions are key factors in the functioning of ecosystems and driving evolutionary processes [87]. A substantial body of research, investigating the impact of fungi as parasites, was conducted on freshwater plankton [88,89]. The ecological impact of fungal pathogens shows that infection caused by zoosporic fungi can be one of the main constraints controlling the calanoids population dynamics in freshwater ecosystems [90,91].

Redfield and Vincent [92] conducted a study on the effects of a fungus belonging to the genus *Lagenidium*, which attacks copepod eggs. The study showed that the infection affected the population dynamics of *Diaptomus novamexicanus* in Castle Lake, California, and suggested that parasitism may exert a greater influence on population regulation than predation, particularly in certain years.

Burns [93] discovered a new fungus, *Aphanomyces* sp., of the family Saprolegniaceae, which parasitized eggs of the copepod *Boeckella dilatata* in Lake Hayes. They examined the growth parameters of this fungus in copepod populations and found that it caused egg and female mortality. The incidence of parasitism was high during the winter and the following summer, leading to reduced birth rates [94]. Again, the presence of fungi played a role in regulating the population growth of copepods. Miao and Nauwerck [95] observed a fungus of the same genus parasitizing the eggs of *Eudiaptomus gracilis* in Mondsee Lake, Austria. They found the fungus infected the eggs throughout the year, significantly reducing female fertility and leading to a decline in zooplankton abundance.

Rossetti [96] examined the parasitism of the fungi Saprolegniaceae on the Copepoda *Eudiaptomus intermedius* populations in the Northern Apennines lakes of Italy. The author presented new data on the dynamics and phenology of the host-parasite interaction across different years and sites. The occurrence of the parasite on a regional scale was assessed, and possible mechanisms involved in pathogen dispersal and host recognition were proposed.

Czeczuga et al. [97] explored the fungal species composition of the planktonic crustaceans. The authors identified 49 species of aquatic fungi in two trophically different lakes; 23 of these grew on copepod carapaces. Czeczuga et al. [98] summarized observations of fungi on dead crustaceans, including copepods, Cladocera, and invertebrates, in six different water bodies, such as springs, rivers, and lakes, and found hundreds of aquatic fungi, including *Chytridiomycetes*, *Hyphochytriomete*, and *Oomycetes*. The number of fungal species varied among different crustacean species, with higher diversity observed on *Daphnia pulex*, *Daphnia magna*, and *Cyclocypris laevis*. The fungal population was found to be correlated with sulfate, calcium density, and chloride in the water column. Through observations of dead crustaceans parasitized by fungi, the fungus *Chytridiales* appeared in large numbers on crustaceans that had been dead for a few days, whereas *Blastocladiales*, *Lagenidiales*, and *Olpidiales* were found on their carapaces a week or two later. Additionally, the identified 21 fungal species were known as fish parasites or necrotrophs. Assessing host susceptibility to fungal infection in relation to the physiological adaptations and genetic characteristics of the microcrustacean populations is of interest. The study highlighted the potential role of dead crustacean specimens as substrates for certain fungal species. These studies highlight the importance of investigating parasite-host interactions and suggest future research in marine ecosystems and in fields of interest such as those related to the biodegradative processes of macromolecules. In fact, fungi play an important role in their degradative activity, especially on plankton carcasses [99]. The exoskeleton of zooplankton, which contains a significant proportion of chitin, provides a suitable substrate for dense colonization by aquatic fungi. Various fungi are typically isolated and cultured with chitin,

and fungal chitinases exhibit effective chitin degradation capabilities [100]. Together with carcasses, the exuviae, resulting from copepod molting, serve as important sources of organic carbon, not only for fungi but in general for a plethora of microorganisms.

### 3.3. Copepod-Virus Interactions

As major pathogenic agents, viruses may regulate the mortality and abundance of copepods. Viral infections can indirectly lead to a decline in copepod populations, thereby influencing the overall structure of the aquatic food web. Despite their abundance and diversity in planktonic samples, our understanding of viruses associated with copepods remains limited [8].

Vermont et al. [101] described the impact of viral infection on *Emiliania huxleyi* coccolithophore on copepod *A. tonsa* ingestion rate. The study showed that during viral infection, phytoplankton physiology and biochemicals changed, and the grazing rate was significantly reduced in copepods. The findings suggest that viral infections can alter food web structure, potentially leading to lower efficiency. The study highlights the importance of considering copepod grazers in understanding the overall impact of coccolithovirus infection on ecosystem function and carbon transfer during *E. huxleyi* blooms.

Thingstad et al. [102] summarized the impact of viruses on the pelagic microbial food web, particularly in Arctic mesocosms, and the challenge of incorporating viruses into dynamic food web models. Viruses are recognized for redirecting material away from the predatory pathway toward detritus and dissolved material, influencing biogeochemical functions. The solution involves introducing adaptations to the defensive and competitive traits of the host community. The study demonstrates how this approach reproduces key aspects of viral dynamics observed in Arctic mesocosm experiments. These experiments link microbial trophodynamics to trophic cascades generated by the seasonal vertical migration of large Arctic copepods. The findings offer a quantitative theory for understanding mechanisms regulating virus-to-prokaryote and lysis-to-predation ratios, emphasizing the central role of predator top-down control in pelagic microbial food webs. The relationship between them is a key factor in maintaining ecological balance, and the regulation of copepod population density by viruses prevents their excessive abundance, thereby sustaining the stability of the ecosystem.

Drake et al. [103] studied the impact of naturally occurring viruses on the copepod *A. tonsa* Dana. The researchers exposed laboratory-reared copepod cultures to elevated concentrations of natural viruses but found no negative effects on copepod fecundity, larval survival, or adult survival. Three possible interpretations were considered: the absence of viruses specific to *A. tonsa* in the concentrated seawater, the presence of pathogenic viruses that did not infect copepods due to various reasons, or latent, lytic, persistent, or tumorous infections that were not evident in measured outcomes. The study emphasizes the need for specific genetic probes to identify viruses infecting *A. tonsa* and confirms the complexity of virus-host interactions.

Dunlap et al. [104] emphasized the crucial importance of copepods in marine ecosystems; however, despite their ecological significance, the causes of copepod mortality remain largely unknown, with up to 35% unaccounted for by predation alone. Dunlap [105] discussed the groundbreaking discovery of viruses in marine mesozooplankton, specifically the copepods *Labidocera aestiva* and *A. tonsa*, in Tampa Bay, Florida. Viral metagenomics revealed two circoviruses, AcCopCV and LaCopCV, in their respective copepod species. LaCopCV was highly prevalent in *L. aestiva*, with active viral replication indicated by transcription. AcCopCV was sporadically detected in *A. tonsa*. Virus-like particles observed under transmission electron microscopy showed active proliferation in copepod connective tissue. This study is the first to describe viruses in copepods and circoviruses in marine invertebrates, addressing a major gap in zooplankton ecology. Advances in high-throughput sequencing are expected to uncover more copepod-associated viral diversity. In addition, environmental stressors also influence the susceptibility of copepods to viral infections.

Variations in temperature and salinity can impact the prevalence and severity of viral infections within copepod populations.

The findings suggest widespread viral presence in marine mesozooplankton, emphasizing the need for further research to understand the ecological impacts of viruses on copepod populations and on the broader mesozooplankton community.

*3.4. Copepod-Protist Interactions*

The interactions between protists and copepods constitute a fundamental component of marine ecosystems. They can either coexist without disturbance or engage in mutually beneficial relationships with them [106]. These intricate interactions involve a variety of ecological and physiological processes, impacting aspects such as nutrient cycling, energy transfer through food webs, and the overall health and stability of aquatic environments.

Protists, as a crucial component of copepod diets, provide essential nutrients for their growth and reproduction [107]. Due to variations in the feeding behaviors and nutrient preferences of different copepod species, they engage in selective grazing, influencing the community composition of protists [106].

Burns and Schallenberg [108] investigated the relationship between copepods and protists, specifically calanoid copepods versus cladocerans, in lakes of varying trophic statuses. Through their consumption of protozoa, these metazooplankton species connect classical food chains and microbial food webs in aquatic ecosystems. The research, conducted in four lakes ranging from ultraoligotrophic to eutrophic conditions, reveals that copepods, particularly calanoid copepods *Boeckella* spp., significantly impede protozoa growth, and their clearance rates are higher in nutrient-poor conditions than in nutrient-rich ones. In oligotrophic lakes, calanoid copepods exhibit higher biomass-specific ingestion rates of heterotrophic nanoflagellates (HNF) compared to the cladoceran *Daphnia*. The study emphasizes that copepods, especially in eutrophic conditions, are more effective consumers of protozoa than cladocerans. It highlights the potential importance of protozoa in the diets of both copepods and cladocerans across lakes with different productivity levels.

The presence of copepods affects the population dynamics of protists, preventing their overgrowth and maintaining balance within microbial communities. The nutrient substances released through copepod excretion can further impact the growth and composition of protist communities. Wickham [109] delved into the trophic relations between cyclopoid copepods and ciliated protists, exploring complex interactions that connect microbial and classic food webs. Through two field experiments, the presence or absence of *Cyclops abyssorum*, *Cyclops kolensis*, and zooplankton larger than 64 μm was manipulated to assess the importance of direct cyclopoid predation on protists versus indirect effects through predation on other metazooplankton. Results showed that the effects of cyclopoids on ciliates depend on predator and prey species, as well as the abundance of alternate prey for cyclopoids. A trophic cascade was observed, particularly for two small ciliates and specifically with *C. abyssorum*. The study suggests that in cyclopoid-ciliate interactions, the switching behavior of the predator may be as crucial as a trophic cascade.

Levinsen et al. [110] explored the trophic coupling between protists and copepods in Arctic marine ecosystems through grazing experiments with *C. finmarchicus*, *Calanus glacialis*, *Calanus hyperboreus*, and *A. longiremis* in Disko Bay, West Greenland, and Young Sound, Greenland. Female copepods, especially during the post-bloom period, exhibited a preference for large protists, notably ciliates and heterotrophic dinoflagellates. Low grazing by *C. glacialis* and *C. hyperboreus* in mid-June suggested a cessation of feeding before overwintering. Clearance rates increased with ciliate and dinoflagellate size, reaching a maximum at 30 to 40 μm equivalent spherical diameter. Notably, *C. finmarchicus* exhibited a lower size limit for capturing *Phaeocystis* single cells (<5 μm) in contrast to *C. glacialis* and *C. hyperboreus*, with a lower limit near 10 μm. The study emphasizes the role of prey and/or predator behavior, in addition to the size and relative concentrations of phytoplankton and heterotrophic protists, in influencing copepod feeding.

**Table 1.** Microbiome associated with copepods.

| Copepod Species | Bacteria/Fungi | Habitat | Citation |
|---|---|---|---|
| *Centropagidae* and *Clausocalanidae* | Bacilli and Actinobacteria | Marine | [38] |
| *Pleuromamma* spp. | Lactobacillales, Bacillales | Marine | [44] |
| *Acartia* sp., *Temora longicornis* | Actinobacteria, Firmicutes | Marine | [46] |
| *Pleuromamma, Undinula* | Vibrionaceae, Oceanospirillales, and Rhodobacteraceae | Marine | [49] |
| *Acartia tonsa* | Unknown | Marine | [64] |
| *Calanus hyperboreus* and *C. glacialis* | Unknown | Marine | [65] |
| *Undinula vulgaris, Pleuromamma* spp. | *Vibrio* spp. | Marine | [66] |
| *Calanus* sp. | Bacteroidetes, Proteobacteria, Actinobacteria | Marine | [67] |
| *Temora stylifera* | *Pseudomonas* | Marine | [68] |
| *Acartia tonsa* | *Bacillus* spp. | Marine | [69] |
| *Acartia bifilosa, Eurytemora affinis* | Deltaproteobacteria, Firmicutes | Marine | [74] |
| *Acartia tonsa* | *Vibrio cholerae* and *Pseudomonas* sp. | Marine | [77] |
| *Lepeophtheirus salmonis* | *Aeromonas salmonicida* | Marine | [111] |
| *Acartia omorii* | Unknown | Marine | [112] |
| *Pleuromamma, Undinula,* and *Sapphirina,* among others | Gammaproteobacterial | Marine | [113] |
| *Acartia tonsa* | *Vibrio cholerae* | Marine | [114] |
| *Acartia* spp. | Unknown | Marine/freshwater | [71] |
| *Mesocyclops aspericornis* | *Wolbachia* | Freshwater | [60] |
| *Diaptomus* spp. | *Pseudomonas, Bacillus, Acinetobacter,* and *Flavobacterium* | Freshwater | [72] |
| *Eurytemora affinis* | *Vibrio* sp. F10 9ZB36 | Freshwater | [83] |
| *Calanus* | *Dikarya* | Marine | [115] |
| *Diaptomus novamexicanus* | *Lagenidium* (Phycomycetes) | Freshwater | [92] |
| *Boeckella dilatate, Diaptomus gracilis* | *Aphanomyces ovidestruens* | Freshwater | [93] |
| *Boeckella dilatata* | *Aphanomyces* sp. | Freshwater | [94] |
| *Eudiaptomus gracilis* | *Aphanomyces* sp. | Freshwater | [95] |
| *Eudiaptomus intermedius* | *Aphanomyces* sp. | Freshwater | [96] |
| Unknown | Chytridiomycetes, Oomycetes, and Peronosporales | Freshwater | [97] |
| *Cyclops fuscus, Cyclops vicinus* | Chytridiomycetes, Hyphochytriomycetes, and Oomycetes | Freshwater | [98] |

## 4. Factors Influencing Copepod-Associated Microbiome

The host species plays a crucial role in shaping the microbial composition [116]; copepods can harbor distinct microbial communities in function of their physiology, behavior, and habitat preferences, such as seawater depth, geographical location, chemical factors such as pH, and seasonal changes. Datta et al. [53] investigated early-stage C5 copepodites of *C. finmarchicus* in Norway during early summer and categorized copepod-associated operational taxonomic units (OTUs) into three groups based on their distribution within individual copepods. The structure of bacterial communities on copepod patches is determined by the core microbiota and is influenced by local ecological selection pressures and copepod-specific physiology, such as feeding behavior and co-colonizing bacteria. The microbial communities associated with copepod eggs, nauplii, copepodites, and adults may differ due to differences in their nutritional requirements and physiological processes. Moisander et al. [39] explored multiple interactions between copepod-associated microbiomes in temperate marine ecosystems during early summer. Increased temperatures cause copepods to molt more frequently [117], as reported above by Holland and Hergenrader [72]. Since bacterial colonization generally occurs after each molt, the bacteria on their body surface also increase. Furthermore, they observed that the relative abundance of copepod-associated microbiomes was high in both seawater and copepods when the local temperature increased [118].

Copepod diet is a crucial factor influencing the microbiome. Copepods can feed on a variety of food sources, including phytoplankton, bacteria, and detritus. Differences in diet composition can lead to variations in the copepod-associated microbial communities.

Studies have shown that bacteria associated with food particles represent a large proportion of the copepod microbiome. Different quality and quantity of food can lead to differences in the number of culturable microorganisms in copepods [64], and their core microbiome varies in carbon utilization and nutrient uptake rates depending on the food. Therefore, changes in food quality during this study period may have also indirectly contributed to this variability.

Slight changes in pH can affect the physiology of plankton, such as the growth and survival of phytoplankton and the egg production and hatching rates of zooplankton [119,120]. Thus, chemical factors such as oxygen and pH may also be responsible for changes in the copepod-associated microbiome. For example, acidic conditions, such as those resulting from ocean acidification, have been found to alter the abundance and diversity of copepod-associated bacteria. Decreased pH levels can lead to shifts in the microbial community structure, with certain taxa becoming more dominant or decreasing in abundance. These changes in the copepod-associated microbiome can have implications for the overall health and functioning of copepods, as well as their interactions with other organisms in the ecosystem. Additionally, pH-induced changes in the copepod-associated microbiome may affect the ability of copepods to cope with environmental stressors and their role in nutrient cycling and ecological processes.

Skovgaard et al. [121] observed that the new bacterium Betaproteobacteria *Delftia* sp. Dominated the microbiome of *Calanus hamatus* when the pH was higher than 8.8, while *Simplicispira* spp. And *Stenotrophomonas* spp. Decreased. In contrast, the copepod microbiomes in low-pH water mainly consisted of the genera *Simplicispira*, *Stenotrophomonas*, and *Staphylococcus*. The discussion of factors influencing copepod-associated microbiomes is critical to elucidating the ecological role of microbes in copepod health, nutrient cycling, and overall ecosystem function.

Some copepods show diurnal vertical migration, which allows for the dispersal of their associated bacteria from the euphotic to mesopelagic layers [51]. Grossart et al. [122] employed stratified migration columns in their research to explore how hitchhiking bacteria disperse vertically while being transported by migrating zooplankton. The experiments revealed that migrating *D. magna* facilitated the transport and release of associated bacteria, with an average dispersal rate of $1.3 \times 10^5$ cells per *Daphnia* per migration cycle for the lake bacterium *Brevundimonas* sp. Similar bidirectional vertical dispersal was observed for two other bacterial species, *Pseudonocardia* sp. And *Pimelobacter* sp., although at lower rates. Field observations in Lake Nehmitz confirmed that diurnally migrating zooplankton acquired different bacterial communities from the hypolimnion and epilimnion during the day and night. These findings demonstrate that hitchhiking on migrating animals, such as zooplankton, can serve as an important mechanism for rapidly relocating microorganisms, including pathogens, allowing them to access otherwise inaccessible resources.

Of concern are the consequences of human activities on changes in copepod microbial communities, such as the extremely far-reaching negative effects of plastic pollution on copepod physiology, which is the most influential issue in marine environmental research. The increasing impact of plastic accumulation in the environment and microplastic particles smaller than 5 mm is a problematic aspect of marine environmental pollution [123]. Their tiny microplastics can become food for zooplankton, thus entering the food chain of marine organisms and affecting entire marine communities [124]. Pollution, eutrophication, climate change, and habitat alteration can alter the composition, diversity, and functioning of copepod-associated microbiomes [59]. For example, chemical pollution from industrial and agricultural sources can introduce contaminants into aquatic ecosystems, leading to shifts in microbial community structure and reduced microbial diversity in copepods. Sewage plays a role in determining copepod microbiomes, particularly in the inshore waters of large cities and farms. The nutrient enrichment of sewage, containing nitrogen (N) and phosphorus (P), can influence the abundance and composition of microorganisms interacting with copepods, potentially altering copepod microbiomes [59]. Eutrophication, resulting from excessive nutrient inputs, can promote the growth of harmful algal blooms, which can

have cascading effects on copepod-associated microbiota. Climate change-induced shifts in temperature, salinity, and pH can also influence the copepod microbiome, potentially disrupting important symbiotic relationships. Furthermore, habitat alteration, such as the construction of dams or the destruction of coastal wetlands, can disrupt copepod populations and their associated microbial communities, with yet unknown effects on the whole ecosystem.

## 5. Biodegradation Role of the Copepod-Associated Microbiome

Zooplankton and microorganisms can release and consume large amounts of particulate matter and dissolve organic compounds, and bacteria can thrive in the microhabitat of zooplankton by obtaining organic and inorganic nutrients. De Corte et al. [67] showed that the zooplankton-associated bacterial assemblage was able to metabolize chitin, taurine, and other organic macromolecules. This association mediates biogeochemical processes in environmental waters through the proliferation of specific bacterial populations.

Wäge et al. [125] found the presence of gut-specific prokaryotic taxa and indicator species of methanogenic pathways in both copepods. The relative abundance of archaea and methanogenic bacteria was examined and showed a high degree of variability among individual copepods, highlighting intra- and interspecific variation in copepod-associated prokaryotic communities. The results reveal that the guts of *Temora* sp. And *Acartia* sp. Have the potential to produce methane in trace amounts.

Gorokhova et al. [74] speculated that it is possible that the copepod gut, fecal pellets, and carcasses contained mercury-methylated bacteria [126]. In a clade-specific quantitative PCR assay of copepods and cladocerans, the authors found that the hgcA gene of the methylation-associated bacterial cluster was carried in both copepods, while it was not found in cladocerans. In contrast, the Hg methylation efficiency of fecal pellets was higher in copepods, and it is hypothesized that endogenous Hg methylation in zooplankton contributes to the regulation of methylmercury in marine fish. Its methylation capacity varied synchronously in the microbiome, and this observation contributes to the dynamics of methylmercury in marine food webs.

Sadaiappan et al. [127] studied five copepod genera, *Acartia* spp., *Calanus* spp., *Centropages* sp., *Pleuromamma* spp., and *Temora* spp., and their associated bacteriobiomes. Their meta-analysis revealed five copepod genera with bacteriobiomes capable of mediating methanogenesis and methane oxidation. Among them, the bacteriobiomes of *Pleuromamma* spp. Had potential genes for methanogenesis and nitrogen fixation, and the bacteriobiomes of *Temora* spp. Were involved in assimilatory sulfate reduction and cyanocobalamin synthesis. The bacteriobiomes of *Pleuromamma* spp. And *Temora* spp. Had potential genes responsible for iron transport. There are also flavobacteria clade members that degrade high-molecular-weight organic matter such as cellulose and chitin, which have symbiotic or parasitic interactions with zooplankton and can use the intestines of copepods as a host environment [128].

## 6. Conclusions and Future Perspectives

Copepods play an essential role in the marine food chain and represent a hotspot of biodiversity, with a microbial abundance much higher than those found in their surrounding aquatic environment. In this review, we report the knowledge gained to date on the copepod-microbiome association and their interactions. Knowledge about microbial diversity and their function with copepods, such as the factors influencing their interactions and their role in copepod growth, aquatic biogeochemical cycles, and the biodegradation of recalcitrant macropolymers, is still limited. Most studies within this field were primarily focused on freshwater copepods, potentially overlooking the unique contributions that associated microbiota make to the ecology of these organisms. What emerges is that the link between copepods and their microbiome is unique in that copepods provide a microenvironment for their bacteria and amplify the influence of microorganisms on vectors of nutrient cycling, aquatic food chains, and whole marine ecosystem dynamics.

On the other hand, the copepod-associated microbiome is currently an emerging research topic of great interest due to the future perspectives that this field can offer in numerous biotechnological fields. The potentiality of microbial diversity needs to be fully investigated, for example, in the field of polymer biodegradation. Bacteria and fungi in the intestinal tract of copepods have been found to degrade organic matter at high molecular weight. They are particularly efficient at secreting enzymes that target specific components of polymers that are recalcitrant to degradation. Some of the bacteria associated with *Calanus* copepods are able to break down complex polysaccharides, such as those of phytoplankton cells. These results encourage more investigations into the degradation activity of recalcitrant macromolecules.

Moreover, few investigations have been conducted into copepod-associated microorganisms responsible for both their pathogenicity and their beneficial role as probiotic factors, which can influence aquaculture activity. Although probiotics have received attention in the aquaculture industry, there has been a marked lack of research on copepod-associated microorganisms with probiotic potential and their impact on aquaculture activities. In the future, exploring the biodiversity of copepod-associated microbiomes will represent a challenge in biotechnological studies, such as the use of molecular sequencing technology, which makes it possible to isolate bacteria and fungi, test their antibiotic or probiotic activity, detect and identify strains against pathogens in aquaculture, and find new frontiers in biodegradation processes.

In conclusion, we encourage further studies to explore the biodiversity and functionality of the copepod-associated microbiome and the potential benefits of those yet-undiscovered microorganisms.

**Author Contributions:** Conceptualization, I.B. and J.F.; Formal Analysis, I.B.; Investigation, L.N., V.V. and Y.Y.; Writing—original draft preparation, J.F.; Writing—review and editing, M.M., S.D.G., L.N. and I.B.; Visualization, J.F.; Funding acquisition, I.B., B.G. and X.Y. All authors have read and agreed to the published version of the manuscript.

**Funding:** This work was funded by the China Scholarship Council as PhD fellowship grant No. 202208330035 to Jiantong Feng and by ISPRA.

**Data Availability Statement:** Not applicable.

**Conflicts of Interest:** The authors declare no conflict of interest.

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
