# Peer review of "Marine Copepods as a Microbiome Hotspot: Revealing Their Interactions and Biotechnological Applications"

_water, doi:10.3390/w15244203_

Round 1

Reviewer 1 Report

Comments and Suggestions for Authors

Review for the paper "Marine copepods as microbiome hotspot: Revealing their interactions and biotechnological applications" by Jiantong Feng, Maurizio Mazzei, Simona Di Gregorio, Luca Niccolini, Valentina Vitiello, Yingying Ye, Baoying Guo, Xiaojun Yan and Isabella Buttino submitted to "Water".

General comment.

Copepods are some of the most abundant marine plankton in the global ocean. Members of the Copepoda species vary in size and have different functional roles. Marine copepods play a critical role as a trophic link in pelagic food webs, and they help control the cycling of organic matter in marine ecosystems by consuming bacteria, microalgae, and detritus. Consequently, copepods facilitate the transfer of nutrients and energy to higher trophic levels. Despite recent advances in the study of copepod biology, many aspects of it still remain unclear. Understanding the microbiome of marine copepods requires further attention. The authors provided a current overview of the microorganisms inhabiting marine copepods, with a focus on bacterial and fungal communities associated with the copepods. This comprehensive review may be of interest to specialists in general marine biology, as well as those working in aquaculture, bioremediation, and biotechnology. With minor revisions, the paper could be accepted for publication.

Specific remarks.

Although the authors focused primarily on bacteria and fungi in copepod microbiomes, it is crucial to note the vital roles of other microorganisms, including marine protists, viruses, and archaea. Two sections briefly describing interactions between viruses-copepods and protists-copepods would enhance the article. Additionally, updating Figure 1 with marine viruses colonizing copepods is recommended. As major pathogenic agents, viruses may regulate the mortality and abundance of copepods.

L33-34. Plankton is the term that typically used in plural. I suggest replacing 'is at the base' with 'are at the base' and 'contributes' with 'contribute'. See also L245.

L61. Artemia should be in Italics according to the Zoological nomenclature.

L476. Aphanomyces ovidestruens should be in Italics.

L483. Consider replacing 'It presents' with 'The author presents'

Table 1. 'sp' and the Latin names of families and higher taxonomical categories must be in ordinary font, not in Italic. Please, correct.

L520. Clarify, which species were considered in the study.

Section 4. The authors must acknowledge the role of sewage in determining copepod microbiomes, particularly in the inshore waters of large cities and farms.

Reference list. It is recommended to carefully verify the Latin names of taxa. In some cases, the species and genera names are not italicized, which should be corrected. L707.

Reference list. Check reference 70.

Comments on the Quality of English Language

Minor revision.

Author Response

Thanks for your reviewing. Please see the attachment.

Reviewer 2 Report

Comments and Suggestions for Authors

The paper will provide a solid introduction to copepod/microbiome interactions for researchers new to the field.

Comments on the Quality of English Language

I have appended a marked manuscript with suggestions for English and clarity improvement.

Author Response

(The authors gave the same response as above.)
